# Language models in molecular discovery

Nikita Janakarajan[1]    Tim Erdmann[2]    Sarath Swaminathan[2]    Teodoro Laino[1]    Jannis Born[1]

[1]IBM Research Europe, Zurich, Switzerland [2]IBM Research Almaden, San Jose, CA, United States
{nja,teo,jab}@zurich.ibm.com
{tim.erdmann,sarath.swaminathan}@ibm.com

## Abstract

The success of language models, especially transformers in natural language processing, has trickled into scientific domains, giving rise to the concept of "scientific language models" that operate on small molecules, proteins or polymers. In chemistry, language models contribute to accelerating the molecule discovery cycle as evidenced by promising recent findings in early-stage drug discovery. In this perspective, we review the role of language models in molecular discovery, underlining their strengths and examining their weaknesses in de novo drug design, property prediction and reaction chemistry. We highlight valuable open-source software assets to lower the entry barrier to the field of scientific language modeling. Furthermore, as a solution to some of the weaknesses we identify, we outline a vision for future molecular design that integrates a chatbot interface with available computational chemistry tools. Our contribution serves as a valuable resource for researchers, chemists, and AI enthusiasts interested in understanding how language models can and will be used to accelerate chemical discovery.

## 1   Introduction

The Turing test – envisioned in 1950 as a machine's ability to simulate human behavior to the extent of indiscernibility – served for decades as the holy grail of artificial intelligence (AI). Recently, language models (LMs) have demonstrated an astonishing ability to understand and generate human-like text [61]. Thanks to this remarkable progress over the last 5-10 years, the perception of the Turing test has undergone a sudden turnaround, shifting from a heavily-debated and largely deemed intractable challenge to a silent, yet widespread acknowledgement of its decipherment. Machine learning (ML) in general and LMs in particular hold the potential to profoundly accelerate the molecular discovery cycle (see Figure 1). Here, we explore applications of LMs to chemical design tasks.

Despite technological advances constantly reshaping our understanding of biochemical processes, the chemical industry persistently faces escalating resource costs of up to 10 years and 3 billion dollars per new market release [96]. The intricacy of the problem is typically attested by an exorbitant attrition rate in *in vitro* screenings [73], the sheer size of the chemical space [63] and the frequency of serendipity [37].

Although LMs were originally developed for natural language, they have shown compelling results in scientific discovery settings when applied to "scientific languages", e.g., in protein folding [51] or *de novo* design of small molecules [99], peptides [21] or polymers [62]. But what exactly is a language model? By definition, it is any ML model that consumes a sequence of text chunks (so-called tokens) and is capable to reason about the content of the sequence. Since each token is essentially a vector [58], a LM is a pseudo-discrete time series model. Most typically, LMs learn probability distributions over sequences of words thus also facilitating the generation of new text given some input, for example in a language translation task. While all LMs rely on neural networks,

NeurIPS 2023 AI for Science Workshop.

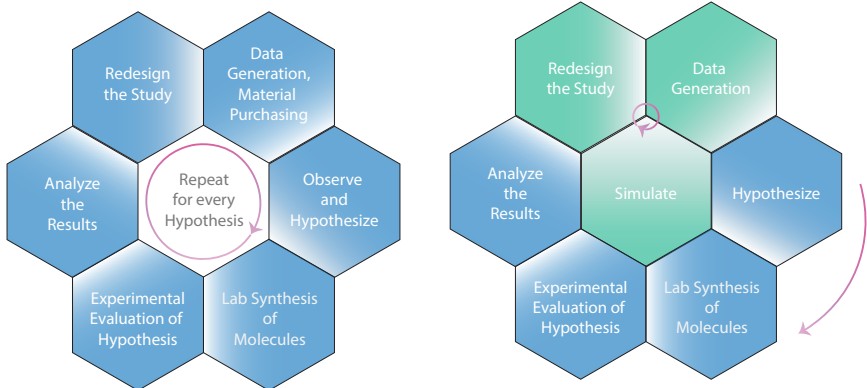

**(a)** Classic molecular discovery.  **(b)** Accelerated molecular discovery.

**Figure 1:** A comparison of molecular discovery workflows: (a) classic approach, where each hypothesis (a.k.a. molecule) requires a new experimental cycle. (b) *Accelerated* molecular discovery cycle with machine-generated hypotheses and assisted validation, enabling simultaneous generation and testing of numerous molecules.

contemporary models almost exclusively leverage the Transformer architecture [87]. Now, all of this begs the question – what is the need for LMs in molecular discovery?

First, when applied to serializations of chemical entities (e.g., SMILES [92]), LMs can learn highly structured representations, often even tailored for desired functional properties [33]. This allows to perform smooth and property-driven exploration of the originally deemed discrete protein or molecular space. Another attractive feature of scientific LMs is their ability to seamlessly bridge natural and scientific languages. This can give rise to ChatGPT-style chatbot interfaces that allow chemists to formulate their design objectives through natural language and to iteratively refine their result with an interactive agent thus potentially accomplishing complex chemical tasks more rapidly. Nevertheless, large-language models (LLMs) like GPT, which power conversational agents, lack knowledge about scientific operations (e.g. molecular discovery), access to information sources providing up-to-date data, and the ability to accurately reference. They tend to hallucinate in their responses, which raises questions about credibility, trust, and applicability. However, this crucial gap between AI and science can be overcome by integrating task-specific agents into the LLM-powered conversational application and allowing the LLM to reason over their appropriate usage based on provided instructions. This also eliminates the application barriers associated with expert-developed AI models, which typically require originating programming and AI/ML skills from the intended user group, often comprising lab scientists. Additionally, it can be anticipated that this will result in a significant increase in the utilization of the developed AI models and contribute to scientific discovery. Here, we present an overview of the role of LMs toward accelerated molecular discovery. We commence with the conventional scientific discovery method and then discuss how molecular generative models can be coupled with molecular property prediction models. Next, we provide readers looking for practical usability with a curated list of software tools and libraries for scientific language modeling. We conclude by envisioning the future of molecule design, where natural language models, custom-built AI models, and cheminformatics tools are integrated into the discovery process via chatbot user interfaces.

## 2 Accelerated molecular discovery

Molecule discovery, intricately linked to optimizing diverse properties in a vast space, challenges conventional scientific methods. In chemistry's Design-Make-Test-Analyze (DMTA) cycle, synthesis costs and time constraints create a bottleneck that hampers hypothesis refinement (cf. Figure 1a). Traditional approaches are largely driven by medicinal chemists who design "molecule hypotheses" which are biased, ad-hoc and non-exhaustive. This hinders progress in addressing global issues, creating a necessity for an accelerated process of molecule discovery. Thus, a key challenge lies in improving the speed and quality of evaluating such "molecule hypotheses", which are grounded on laboratory work.

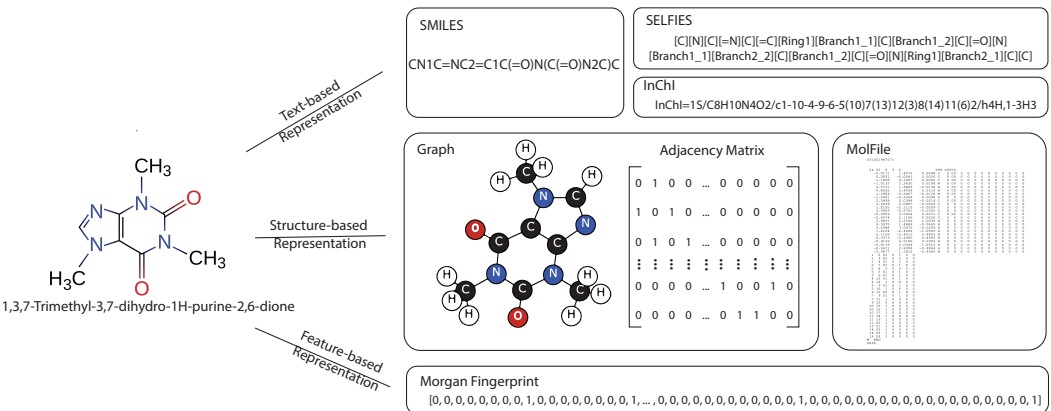

**Figure 2:** An illustration of popular ways of representing a chemical molecule as input to a ML model. The representations may be (a) String-based, such as SMILES, SELFIES, or InChI which use characters to represent different aspects of a molecule, (b) Structure-based, such as Graphs or MolFiles that encode connectivity and atomic position, and (c) Feature-based, such as Morgan Fingerprints, which encode local substructures as bits.

Deep generative models have recently emerged as a promising tool to expedite the hypothesis/design phase in molecular discovery. However, even the most advanced molecular generative models require an efficient method for large-scale virtual screening to test their hypotheses. The *accelerated molecular discovery* cycle adds a validation loop to DMTA, rapidly evaluating numerous hypotheses inexpensively (cf. Figure 1b). This loop enhances the design-phase generative model, ensuring only promising hypotheses advance to the synthesis and physical experimentation stages.

## 2.1 Molecule Representation

Data representation plays a crucial role in molecular discovery. It determines the type of information that is available to the model and consequently the properties that can be predicted. An overview of commonly used molecular representations for property prediction is illustrated in Figure 2. Due to the popularity of chemical language models (CLMs), this section focuses on text-representations of molecules. A more focused discussion on CLMs is covered by Grisoni [35].

**Simplified Molecular Input Line-Entry System (SMILES)**    SMILES [92] is a string representation made up of specific characters for atoms, bonds, branches, aromaticity, rings and stereochemistry in molecular structures. The character-level representation enables easy tokenization, making SMILES an ideal input for LMs. SMILES are typically tokenized at the atom level [75, 86]. For LMs to learn from SMILES, tokens are typically vectorized either via one-hot encodings (where each row in the binary matrix corresponds to a SMILES position and each column signifies a token) or by learning a continuous embedding for each token during training.

**Self Referencing Embedded Strings (SELFIES)**    SELFIES [46] were introduced as an alternative to SMILES to counter the problem of generating invalid molecules. Unlike SMILES, SELFIES are generated using derivation rules to enforce valence-bond validity, and additonally store branch length and ring size.

**International Chemical Identifier (InChI)**    Introduced by the IUPAC, InChI [38] are strings encoding structural information including charge of the molecule in a hierarchical manner. InChIs are less commonly used in LMs [36].

## 2.2 Generative Modelling

Generative modeling involves learning the data's underlying distribution with the intent of generating new samples, a technique pivotal in accelerating de novo drug discovery. A generative model may be conditional or unconditional. A conditional generative model utilizes provided data attributes or labels to generate new samples with desired properties, whereas an unconditional model solely

provides a way to sample molecules similar to the training data [33]. The DMTA cycle particularly benefits from the conditional generation approach as it facilitates goal-oriented hypothesis design [7]. This section describes a few influential conditional generation models that act on chemical language to generate molecules satisfying user-defined conditions.

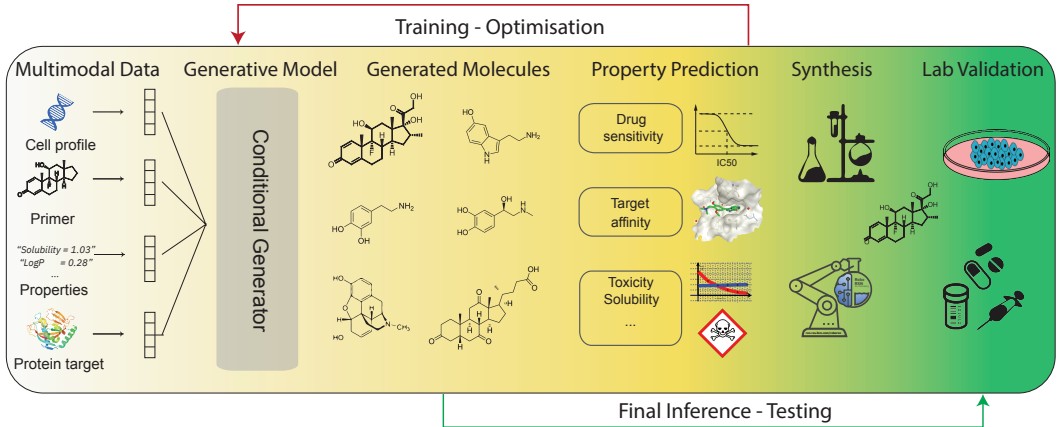

**Figure 3:** An illustration of conditional molecule generation using LMs. The process initiates with the collection and processing of multi-modal data, which is then compressed into a fixed-size latent representation. These representations are subsequently passed to a molecular generative model. The generated molecules then undergo in-silico property prediction, which is linked back to the generative model through a feedback loop during training. The in-silico models direct the generative model to produce property- or task-driven molecules using a reward function. In the inference stage, candidate molecules generated by the optimized model undergo lab synthesis and subsequent experimental validation to determine their efficacy for the desired task.

**Recurrent Neural Network (RNN)**    The sequential nature of RNNs makes them suitable models for processing chemical languages. Proposed in the 90s, RNNs were the first type of CLMs to enter the domain [6, 75, 81]. RNNs continuously update their hidden states as new tokens are passed to the network, thus enabling it to encode contextual information. During the generation process, tokens are produced auto-regressively. RNNs find use in generating molecule libraries [81] which are extensively used in drug development processes like screening. External scoring functions drive the generation of molecules with desired properties. RNNs are also adept at learning complex distributions [28] and generating a higher proportion of unique and valid SMILES [64], even though their inability to count occurrences of ring opening/closing symbols poses a challenge [43, 65].

**Variational Autoencoder (VAE)**    VAEs learn latent distribution parameters of molecules, thus enabling the generation of new molecules by sampling from this distribution. Their unique ability lies in learning a smooth, latent space that facilitates interpolation of samples, even for notoriously discrete entities like molecules [33]. To make it suitable for chemical language models (CLMs), any network compatible with string inputs can function as a VAE's encoder and decoder. Initial works primarily focused on single-modality applications, assessing latent space quality via downstream tasks [33]. This approach remains prevalent and can be used to generate, e.g., catalysts with an RNN-based VAE [74] . Here, a latent space is learned and assessed by predicting the catalyst binding energy. Lim et al. [49] takes it a step further by concatenating a condition vector to the input and the latent embedding generated by the recurrent network-based VAE's encoder. This approach enables the generation of molecules specifically tailored to the given conditions. The scope of VAEs expanded progressively into multi-modal settings for conditional molecule generation, as visualized in Figure 3 and exemplified by Born et al. [9, 10, 11]. These works on task-driven molecule generation incorporate contextual information like gene expression [11] or protein targets [9, 10] or even both [42]. VAEs learn embeddings of context information and primer drugs, which are merged before decoding to produce molecules. A reinforcement-learning-based approach directs the model to produce molecules with desired properties using rewards.

**Transformer**    The self-attention attribute of Transformers [87] have propelled these models to the forefront of NLP. Transformers have an encoder module that relies on this self-attention to learn

embeddings of the input and the context associated with this input. The decoder module predicts tokens using the context learnt by the encoder and previously generated tokens through attention. For generative modeling, decoder-only transformer like the Generative Pre-Training Transformer (GPT) [67] have become the dominant approach. This success was translated to the scientific language domain. One of the first models to use the GPT architecture for conditional molecule generation is MolGPT [3]. SMILES tokens concatenated with a condition vector that summarizes the desired properties and scaffolds are passed as input to this model, which is then trained on the next token prediction task to generate molecules. GPT-like models coupled with RL can also be used to optimize molecular properties like pIC50 [57]. In this two-stage approach, embeddings are first learnt from SMILES strings, and the embedding space is then optimized such that the model samples molecules with the desired properties. Going beyond just using GPT-like architectures for molecule generation, Regression Transformer [8] is a seminal work that formulates conditional sequence modeling as a regression problem. This gives rise to a natural multitask model that concurrently performs property prediction and conditional molecular generation. This is achieved by concatenating conventional molecular tokens with property tokens and employing an training scheme that alternates which parts of the sequence are masked.

The superior quality of learned embeddings coupled with its ability to handle parallel processing and scalability makes the Transformer a top choice for the task of conditional molecule generation, with promising applications in drug discovery and other areas of molecular design [62].

### 2.3 Property Prediction

Property prediction is a key step in validating the molecules for a given use case. The success of a molecule depends on a myriad of factors, including how it interacts with its environment. The MoleculeNet datasets [97] are a commonly used benchmark for property prediction. It is curated from public datasets and comprises over 700,000 compounds tested on various properties. A recent trend is to use transformer-encoders to learn embeddings for molecules and then apply a multilayer perceptron (MLP) on the embeddings for property prediction. MolBERT [26] and ChemBERTA [18]) are two such examples. These transformer-based models use a BERT backbone to learn molecular embeddings from SMILES and predict properties. Similarly, Molformer [71] uses a transformer-encoder with linear attention and relative positional encoding to learn compressed molecular representations which are then fine-tuned on chemical property prediction benchmarks. To equip transformers with better inductive biases to handle molecules, adaptations of the attention mechanism were proposed. The molecule attention transformer (MAT) incorporates inter-atomic distances and graph structure into the attention mechanism [54]. An improvement over this model is the *relative*-MAT which fuses the distance embedding, bond embedding and neighbourhood embedding and achieves competitive performances on a range of property prediction tasks [55].

## 3 Software tools for scientific language modeling

The paradigm shift towards open-sourcing software has profoundly influenced chemistry. Commonly listed implications of open-sourcing in the context of drug discovery include catalyzation of methodological development, fostering collaboration and ease of scientific reproducibility [32]. In this section we present several software assets (e.g., Python packages or cloud-based web apps) that are key to enable molecular discovery.

**Natural language models** The success story of the Transformer [87] as most widely adopted neural network architecture goes hand in hand with the rise of the `transformers` library [95]. Initially intended for NLP applications, Transformers were adopted across disciplines, e.g in computer vision [22], reinforcement learning [17], protein folding [44] and naturally, chemistry [80]. *Hugging-Face* provides the largest public hub of LMs and it offers implementations of all recent models as well as a diverse collection of pretrained models available for fine-tuning or inference. While most of their models focus on NLP, select models are designed for life science applications, in particular, molecular property prediction (e.g., *ChemBerta* [18]), molecular captioning (e.g., *MolT5* [23]), text-based molecular generation (e.g., *MolT5* [23]) and unsupervised protein language modeling (e.g., *Prot-Bert*, *ProtAlbert*, *ProtXLNet* and *ProtT5* [24]). Furthermore, some available models like *Multimodal Text and Chemistry T5* [20] are prompt-based multi-taskers that extend beyond the above mentioned tasks to include additional functions like predicting forward/backward reactions.

**GT4SD – Generative Toolkit for Scientific Discovery**    Python libraries like `gt4sd` [53]), TdC (Therapeutics Data Commons [40]) or `deepchem` [68] were developed primarily for molecular discovery applications. `gt4sd` in particular provides extensive support for CLMs. GT4SD is designed to enable researchers and developers to use, train, fine-tune and distribute state-of-the-art generative models for sciences with a focus on organic materials. It is compatible and inter-operable with many existing libraries such as `transformers,diffusers` [90], `torchdrug` [100]) or `tape` [69]. Besides established benchmarks for molecular generation, such as Moses [64] and GuacaMol [14] which includes VAEs, generative adversarial networks (GANs), genetic algorithms, and many evaluation metrics for molecular design, `gt4sd` also provides supports for contemporary models like the *Regression Transformer* for concurrent sequence regression and property-driven molecular design [8], *GFlowNets* for highly diverse candidate generation [4] and *MoLeR* for motif-constrained molecule generation [56]. Models can be trained through a CLI with a few lines of code and can be shared to a cloud-hosted model hub. It is built to facilitate consumption by containerization or distributed computing systems,includes $\sim 50$ property prediction endpoints for small molecules, proteins and crystals, and overall hosts $\sim 30$ pre-trained algorithms for material design, 20 free webapps [2] and many Jupyter/Colab notebooks.

**RXN for Chemistry: Reaction and synthesis language models**    Once a molecule has been selected for experimental validation, a tangible synthesis route has to be identified. Since the most important tasks in chemical reaction modeling can be framed as sequence conversion problems, the methodology developed for natural language translation can be seamlessly translated to chemistry [80]. The most mature and flexible library for reaction modeling with LMs is the package `rxn4chemistry` [29]. It wraps the API of the *IBM RXN for Chemistry* platform, a freely accessible webapp that gives access to a rich set of CLMs for different tasks in reaction chemistry. The primary architecture is the *Molecular Transformer* (MT), an autoregressive encoder-decoder model, originally applied to predict outcomes of chemical reactions in organic chemistry [76]. The MT was applied to single-step retrosynthesis [85] and became vital to multi-step retrosynthesis model with a hypergraph exploration strategy [77]. This approach was later generalized to enzymatic reactions with a tokenization scheme based on enzyme classes, facilitating biocatalyzed synthesis planning, and paving the road towards greener chemistry [66]. Derivatives of the MT helped to enhance diversity in single-step retrosynthesis [85] and a prompt-based disconnection scheme improved controllability by allowing the user to mark a disconnection side in the reactant [84]. Interestingly, an encoder-only derivative of the MT excelled in predicting reaction classes [79]. Its hidden representations were found to encode reaction types thus allowing to map reaction atlases and to perform reaction similarity search through the `rxnfp` package for reaction fingerprinting. Strikingly, this led to the discovery that the learned attention weights of the Transformer are "secretly" performing atom mapping between products and reactions [78].

Once the precursors for a synthesis route are identified, the subsequent phase seeks for an actionable, stepwise synthesis protocol that is ideally amenable for autonomous execution on a robotic platform, such as *IBM RoboRXN*. In two seminal works Vaucher et al. demonstrated that encoder-decoder Transformers can extract chemical synthesis actions, first from experimental procedures described in patents [88] and later predict them directly from the reaction SMILES [89]. These models are available via the *IBM RXN for Chemistry* platform which even allows to control and monitor the robotic platform directly from the web interface. For multistep retrosynthesis, *RXN* also includes other models like *AiZynthFinder* [31], a Monte Carlo Tree Search approach build on top of a RNN.

**Specialized libraries**    RDKit [47] remains the best and often only library for manipulating molecules in Python. `HuggingMolecules` is a library solely devoted to aggregating, standardizing and distributing molecular property prediction CLMs [30]. It contains many encoder-only models, some of them with geometrical and structure-aware inductive biases (e.g., the MAT [54] or its successor, the R-MAT [55]) while others are pure BERT-based models that were trained on SMILES (e.g,. *Mol-BERT* [26] or *ChemBERTA* [18]). For narrower applications, like ML data preparation, several tools exist. First, `rxn-chemutils` is a library with chemistry-related utilities from RXN for Chemistry. It includes functionalities for standardizing SMILES (e.g., canonicalization or sanitization) but also conversions to other representations (e.g., InChI). It harmonizes reaction SMILES and prepares them for consumption by CLMs, including also SMILES augmentation and tokenization. Another library with a similar focus is `pytoda` [10, 11]. It does not support reaction SMILES but implements richer preprocessing utilities, allowing to chain >10 SMILES transformations (e.g., kekulization [13]). It supports different languages (e.g., SELFIES [46] or BigSMILES [50]) and tokenization schemes (e.g.,

SMILES-PE [48]). Similar functionalities are available for proteins including different languages (IUPAC, UniRep or Blosum62) and protein sequence augmentation strategies [12].

MELLODDY [39] is a collaborative effort aimed at cross-pharma federated learning (i.e., preserving privacy through decentralized, distributed training) of 2.6 billion confidential activity data points. Similarly, VirtualFlow [34] is an open-source platform facilitating large-scale virtual screening that was shown to identify potent KEAP1 inhibitors. With a focus on *de novo* drug design, Chemistry42 [41] is a proprietary platform integrating AI with computational and medicinal chemistry techniques.

## 4 Future of molecular discovery

A few years ago, the idea of querying an AI model – like one would a search engine – to not only extract scientific knowledge but also perform computational analyses was an overly ambitious feat. Scientific thinking comes from the ability to reason, and AI models cannot reason like humans, yet. However, these models can **learn** from humans. Our propensity to document everything has enabled us to train Large Language Models (LLMs), like ChatGPT [60] and GitHub Copilot [1], to mimic human responses. When brought into the context of computational science, this could equip non-experts to confidently conduct computational analyses through well-designed prompts. With human-in-the-loop, a synergistic effect could be created where the scientist provides feedback to the model on its output, thus aiding in better model optimization (a strategy called reinforcement learning from human feedback (RLHF) that has been proven critical for ChatGPT [19]). These applications also reduce the barrier for individuals from non-scientific backgrounds to gain a more hands-on experience in conducting scientific analyses without having to go through formal training in computational analysis.

This section provides a sneak peak into what's next for molecular discovery. Riding the LLM wave, the future holds a place for chatbot-like interfaces that may take care of all things computational, comprising generating and improving design ideas, synthesis planning, purchasing, and validation.

### 4.1 The rise of foundation models in chemistry

Conventionally, neural networks are trained for a single given task to achieve maximum performance. This essentially renders the models useless for other tasks, thus requiring a new model for every new task, even when the training domain is the same, which in turn imposes a constraint on the rate of our technological advancements. Over the last few years, this conventional approach has been challenged by Large Language Models (LLMs). It has been found that scaling up LLMs leads to astonishing performance in few-shot [15] and even zero-shot task generalization [72]. Referred to as "foundation models" [27, 59], these models, with typically billions of parameters, can perform multiple tasks despite being trained on one large dataset. Essentially, this multi-task learning is achieved by prompting LLMs with task instructions along with the actual query text which has been found to induce exceptional performance in natural language inference and sentence completion [72]. These findings have kicked off new research directions, such as prompt engineering [91] and in-context learning [15], in NLP.

Foundation models find an increasing adoption in chemistry with an increase in task-specific models integrating natural and chemical languages [23, 88, 89, 98]. Concurrently, multi-tasking in pure CLMs has also been advancing through models that combined tasks such as property prediction, reaction prediction and molecule generation either with small task-specific heads (e.g., T5Chem [52]) or via mask infilling (e.g., Regression Transformer [8]). Christofidellis et al. [20] were the first to bridge the gap and develop a fully prompt-based multi-task chemical and natural language model. Despite only 250M parameters, the *Multitask Text and Chemistry T5* was shown to outperform ChatGPT [60] and Galactica [83] on a contrived discovery workflow for re-discovering a common herbicide (natural text → new molecule → synthesis route → synthesis execution protocol).

### 4.2 The coalescence of chatbots with chemistry tools

Given the aforementioned strong task generalization performances of LLMs, building chatbot interfaces around it was a natural next step and thus next to ChatGPT [60], many similar tools were launched. Such tools were found to perform well on simplistic chemistry tasks [16, 93], opening potential to reshape how chemists interact with chemical data, enabling intuitive access to complex

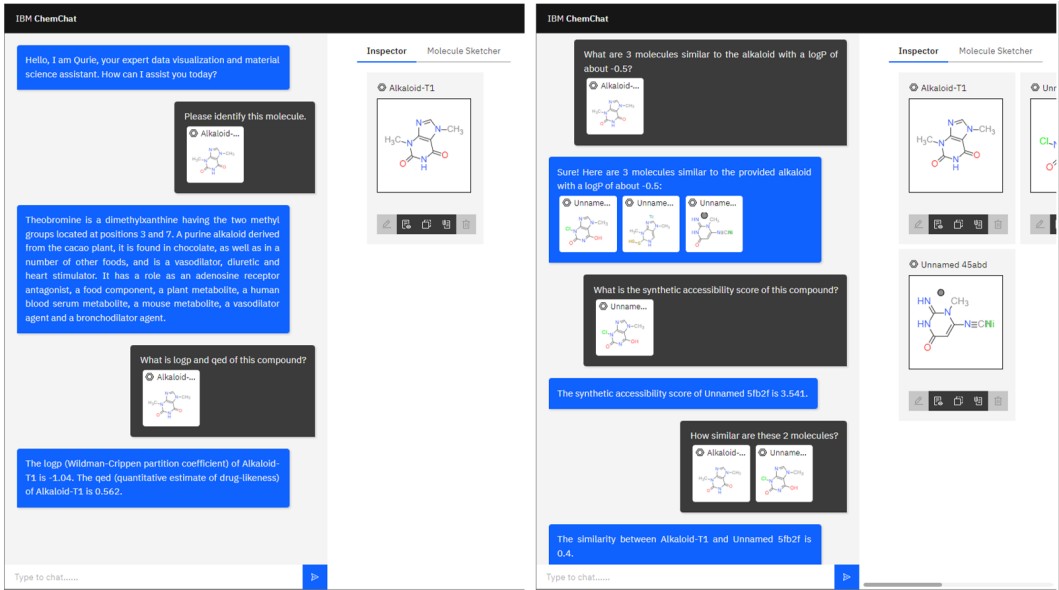

**Figure 4:** Screenshots of the LLM-powered chatbot application `ChemChat`. Embedding the capabilities of existing resources such as identification through PubChem [45], property and similarity calculation through RDKit [25, 47, 70, 82] or generative tasks through GT4SD's Regression Transformer [8, 53] enables the assistant to execute programming routines in the background and, thus, to answer highly subject-matter specific requests without the need for programming skills by the user.

concepts and make valuable suggestions for diverse chemical tasks. Furthermore, AI models specifically developed by computer scientists for e.g. drug discovery or material science can be made available through applications powered by LLMs, such as chatbots. This minimizes the access barrier for subject matter experts who would otherwise require the respective programming skills to utilize these AI models. The power of such chatbots is reached through the coalscence of LLMs and existing chemistry software tools like PubChem [45], RDKit [47] or GT4SD [53]. Together, such applications can unleash the full potential and value of these models by the strongly enhanced usage. An example of how the interaction with such a tool could look like is shown in Figure 4.

In this example, a user provides a molecule and requests identification. The chatbot relies on prompt-engineering in order to inform the LLM about all its available tools. The user input is first sent to the LLM which recognizes that one of its supported tools, in this case PubChem, can answer the question. The application then sends a request to the PubChem API and returns a concise description of the molecule. The user subsequently asks to compute the logP partition coefficient [94] and the quantitative estimate of drug-likeness (QED) [5]. Calculation of both properties is enabled through GT4SD [53] and will trigger a programming routine to accurately format the API request to the instance of GT4SD. The post-processing routine formats the LLM-generated string reply and composes the response object for the frontend. This fusion of LLMs with existing tools gives rise to a chatbot assistant for material science and data visualization that can perform simple programming routines without requiring the user to know programming or have access to compute resources.

A continuation of the conversation involving more complex user queries follows with a generative task (Figure 4, right). Having identified the initial molecule as theobromine with a logP of -1.04, the user requests three similar molecules with a slightly increased logP of -0.5. Here, `ChemChat` identifies the Regression Transformer [8] as the available tool to perform substructure-constrained, property-driven molecule design. Once the routine has been executed and three candidate SMILES are collected, the response is enriched by data of the generated molecules.

In conclusion, chatbots can facilitate the integration of essentially all major cheminformatics software in a truly harmonized and seamless manner. While LLMs are not intrinsically capable to perform complex routines, at least not with high precision and in a trustworthy manner, the synergy between their natural language abilities and existing chemistry tools has the potential to transform the way molecular discovery is performed.

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
