# OpenReview forum: "Language Models in Molecular Discovery"
_NeurIPS.cc/2023/Workshop/AI4Science — NeurIPS2023-AI4Science Poster_

### Official Review · Reviewer_3Usu · 2023-10-07

**Rating:** 9
**Confidence:** 5

**Review:**

This manuscript is a review with high clarity that discusses the application of language models in many sub-fields of molecular discovery. It is comprehensive as it starts from molecule representations and background information of generative models. The manuscript then enumerated some current models for molecular property prediction and RXN planning/prediction. The software packages included in this manuscript can be very helpful for readers to familiarize themselves with this field. Overall, I believe this is a high quality manuscript for the attention track.

---

### Official Review · Reviewer_25Um · 2023-10-08

**Rating:** 2
**Confidence:** 5

**Review:**

The paper titled "Large Language Models in Molecular Discovery" explores the application of language models, particularly transformers, in the field of computational chemistry and molecular discovery.

This is not a coherent and an easy-to-follow paper.

The authors first start with the illustration of Turing test, and suddenly transition into the general description of ML, and finally “explore applications of LMs to chemical design tasks ”

These parts are completely disjoint and do not have a logical connection.

Then the authors continue to talk about the issues of up-to-date data and hallucination in large models. None of these points are illustrated in details.

Similarly confusing is the illustration of section 2. the authors illustrates RNN, VAE, transformer without building a logical connection between any of those.

There are typos like “additonally“ on the 3rd page. Moreover, the footnote on the first page is wrong. It should not be “NeurIPS 2021”